# Safranal Induces Vasorelaxation by Inhibiting Ca^2+^ Influx and Na^+^/Ca^2+^ Exchanger in Isolated Rat Aortic Rings

**DOI:** 10.3390/molecules27134228

**Published:** 2022-06-30

**Authors:** Noor Nadhim Al-Saigh, Shtaywy Abdalla

**Affiliations:** 1Department of Medical Laboratory Sciences, Faculty of Allied Medical Sciences, Al-Ahliyya Amman University, Amman 19328, Jordan; noor_nadhem_2000@yahoo.com; 2Department of Biological Sciences, School of Science, The University of Jordan, Amman 11942, Jordan

**Keywords:** Ca^2+^ influx, Na^+^/Ca^2+^ exchanger, ouabain, safranal, vasodilation

## Abstract

**Introduction:** Safranal, which endows saffron its unique aroma, causes vasodilatation and has a hypotensive effect in animal studies, but the mechanisms of these effects are unknown. In this study, we investigated the mechanisms of safranal vasodilation. **Methods:** Isolated rat endothelium-intact or -denuded aortic rings were precontracted with phenylephrine and then relaxed with safranal. To further assess the involvement of nitric oxide, prostaglandins, guanylate cyclase, and phospholipase A_2_ in safranal-induced vasodilation, aortic rings were preincubated with L-NAME, indomethacin, methylene blue, or quinacrine, respectively, then precontracted with phenylephrine, and safranal concentration–response curves were established. To explore the effects of safranal on Ca^2+^ influx, phenylephrine and CaCl_2_ concentration–response curves were established in the presence of safranal. Furthermore, the effect of safranal on aortic rings in the presence of ouabain, a Na^+^-K^+^ ATPase inhibitor, was studied to explore the contribution of Na^+^/Ca^2+^ exchanger to this vasodilation. **Results:** Safranal caused vasodilation in endothelium-intact and endothelium-denuded aortic rings. The vasodilation was not eliminated by pretreatment with L-NAME, indomethacin, methylene blue, or quinacrine, indicating the lack of a role for NO/cGMP. Safranal significantly inhibited the maximum contractions induced by phenylephrine, or by CaCl_2_ in Ca^2+^-free depolarizing buffer. Safranal also relaxed contractions induced by ouabain, but pretreatment with safranal totally abolished the development of ouabain contractions. **Discussion/Conclusion:** Inhibition of Na^+^-K^+^ ATPase by ouabain leads to the accumulation of Na^+^ intracellularly, forcing the Na^+^/Ca^2+^ exchanger to work in reverse mode, thus causing a contraction. Inhibition of the development of this contraction by preincubation with safranal indicates that safranal inhibited the Na^+^/Ca^2+^ exchanger. We conclude that safranal vasodilation is mediated by the inhibition of calcium influx from extracellular space through L-type Ca^2+^ channels and by the inhibition of the Na^+^/Ca^2+^ exchanger.

## 1. Introduction

*Crocus sativus* L., commonly known as saffron, is a member of the Iridaceae family. The perennial stemless herb has been used as a spice and as a medicinal plant in Mediterranean countries since ancient times. The major components of saffron include crocin, picrocrocin, and safranal [1]. These ingredients are found in crocus in significant quantities, especially in the dried red stigmas of the flower [2], although flavonoids and other pigments are also found in small quantities. Nevertheless, crocin and safranal are the main components of saffron which are responsible for its biological effects [3]. Crocin is responsible for the color of saffron; picrocrocin, a precursor of safranal, gives it the bitter taste; and the volatile agent safranal provides the aroma [4,5], making it the world’s most expensive spice [2].

The active components of saffron, like many other plants, have recently been used as efficient and safe alternative natural drugs with cultural acceptability and fewer side effects [4]. These ingredients have demonstrated a wide spectrum of biological effects, such as anti-oxidant [6], analgesic, and anti-inflammation [7]. Other effects include anti-convulsant [8], anti-anxiety, aphrodisiac [9], anti-depressant [10], anti-tumor [11], bronchodilator, and cardiovascular properties [1,12,13].

Among the pharmacologically active components of saffron is the monoterpene aldehyde safranal (2, 6, 6-trimethyl-1,3-cyclohexadiene-1-carboxaldehyde; Appendix A). Safranal is abundant in the essential oil of saffron since it accounts for 60–70% of the volatile fraction [5,14,15]. It is well known for its many biological activities. These activities (reviewed by Rahmani, et al. (2017)) include an anti-oxidant and an anti-convulsant activity, as well as anti-depressant, anti-anxiety, and hypnotic actions. Further activities include anti-tussive, anti-ischemia, anti-cancer, anti-epileptic, immunomodulatory, pain killing, and tissue protective properties, as well as anti-microbial effects [1,16]. Safranal also has protective action in the cardiovascular system, as demonstrated in animal studies showing that safranal decreased contractility of the myocardium by blocking the calcium channel in guinea pigs [17] and protected against cardiac infarction induced by isoproterenol in rats by decreasing lactate dehydrogenase and creatine kinase of muscle and brain and peroxidation of myocardial lipid [18].

Safranal’s hypotensive effect has been demonstrated in both anaesthetized normotensive and hypertensive rats. Razavi and coworkers showed that safranal reduced systolic blood pressure in rats treated chronically with desoxycorticosterone acetate (DOCA) [17]. Imenshahidi et al. (2013) showed that injections of safranal for 5 weeks in DOCA-salt hypertensive rats and normotensive rats reduced mean blood pressure in the former but did not affect the latter rats [13]. Moreover, Shafiee et al. (2017) demonstrated that intravenous injection of safranal reduced arterial blood pressure in normotensive and DOCA-induced hypertensive rats dose-dependently. [19].

In normal physiological conditions, the endothelium regulates vascular wall homeostasis and the basal vessel tone by the production of vasoconstrictor substances such as endothelins, thromboxane A2, and prostanoids, as well as reactive oxygen species or by the release of vasodilator substances including nitric oxide, prostacyclin, and endothelium-derived hyperpolarizing factor [4,17]. According to Godo and Shimokawa (2017), the initial step toward cardiovascular disease is a dysfunctional endothelium [20]. Therefore, research on safranal has focused on the possible involvement of the endothelium and whether it participates in its hypotensive effects. For example, the relaxant effect of safranal in rat isolated aorta was found to be endothelium-independent, since it was resistant to L-NAME and to indomethacin [17]. These, and other observations [5,21], provide a significant understanding of the smooth muscle relaxant mechanism of safranal, although this understanding requires further investigation [17,19,21]. This work confirms the lack of endothelium interference in safranal-induced vasodilation and describes different mechanisms of action for the most expensive flavoring agent safranal [2]. We show that safranal inhibited Ca^2+^ influx through L-type Ca^2+^ channels and the Na^+^/Ca^2+^ exchanger. In the presence of ouabain to induce the backward mode of the sodium–calcium exchanger, safranal prevents the entry of Ca^2+^ in exchange for sodium ions.

## 2. Materials and Methods

Safranal was dissolved in a double volume of Tween-80, and the volume was diluted with 0.9% NaCl to a final concentration of 10^−1^ M safranal, and dilutions were made with 0.9% NaCl solution so that the final percentage of Tween-80 was less than 0.9% *v*/*v*. Control aortic rings were treated with 0.9% *v*/*v* Tween-80 in physiological salt solution.

### 2.1. Animals

Adult male rats weighing 200–250 g were kept at a temperature of 23 ± 1 °C and with a 12 h light/dark cycle with free access to food and water before the experiments. All the experiments were carried out after the approval of IRB, University of Jordan, decision # 7-2019, dated 25/7/2019.

### 2.2. Tissue Preparation 

The chest of phenobarbital-euthanized animals was opened, and the aorta was carefully dissected and immersed in Krebs bicarbonate buffer aerated with 95% O_2_ and 5% CO_2_. Blood, adherent fat, and connective tissues were cautiously dissected out and the artery was divided into rings 4–5 mm in length. The rings were rapidly immersed in organ baths containing Krebs bicarbonate buffer bubbled at 37 °C with 95% O_2_ and 5% CO_2_ to provide a pH of 7.3–7.4. Krebs bicarbonate buffer was composed of (mM) NaCl 118, KCl 4.7, CaCl_2_. 2H_2_O 2.5, MgCl_2_. 6H_2_O 0.5, NaH_2_PO_4_ 1.0, NaHCO_3_ 25.0, and glucose 11.1. 

Aorta rings were mounted using two stainless steel hooks threaded carefully through the lumen of each ring to protect against endothelial injury. One hook was connected by a thread to a PanLab force transducer (Australia) to record isometric tension in the aortic rings, and the other was fixed to the bottom of the tissue bath. Rings were mounted under a resting tension of 1 g and allowed a period of equilibration of 90 min, during which Krebs buffer was replaced with fresh buffer every 15 min. Responses of aortic rings were recorded using Power Lab (AD Instrument) powered by Lab Chart 7.3 (AD Instruments, Castle Hill, Australia) software [22]. In all the experiments, an aortic ring was used only for one trial.

### 2.3. Role of Endothelium in Safranal-Induced Vasodilation

Aortic rings were precontracted using 5 × 10^−5^ M phenylephrine (Phe), which gave about 90% of the maximum contraction to BaCl_2_. When the contraction reached a steady plateau (35–40 min), cumulative concentration–response curves to safranal (3 × 10^−5^–3 × 10^−3^ M) were established on endothelium-denuded and endothelium-intact aortic rings. The range of safranal concentrations was chosen based on preliminary experiments. For the endothelium-intact experiments, the endothelial layer’s integrity was assessed by relaxation to concentrations of acetylcholine (10^−5^–2 × 10^−4^ M), equal to 76.2 ± 6.0% of phenylephrine contraction. For the denudation experiments, endothelium was mechanically removed by gentle rubbing of the lumen using a fine, smooth wooden toothpick. Endothelial removal was assessed by recording their relaxation in response to acetylcholine, which was equal to 27.4 ± 5.2% of phenylephrine contraction [23]. More rigorous rubbing to ensure complete removal of endothelium was found to decrease tension development since it may damage the smooth muscle cells. 

### 2.4. Role of Nitric Oxide, Prostaglandins, Guanylate Cyclase, or Phospholipase A_2_ in Safranal-Induced Vasodilation 

Endothelium-intact rings were preincubated for 30 min with either 10^−4^ M N^ω^-nitro-L-arginine methyl ester (L-NAME; Sigma-Aldrich, Steinheim, Germany), a nitric oxide synthase (NOS) inhibitor [24]; 10^−5^ M indomethacin (Sigma-Aldrich, St. Louis, MI, USA), a cyclooxygenase inhibitor; 10^−5^ M methylene blue, a guanylate cyclase inhibitor [23,25]; or 10^−6^ and 10^−5^ M quinacrine (Fluka BioChemika, Buchs, Switzerland), a phospholipase A_2_ (PLA_2_) inhibitor [26]. Safranal’s effect was determined in the absence and the presence of these inhibitors. In each set of experiments, safranal was added to the bath after the maximum contraction to Phe 5 × 10^−5^ reached a plateau (35–40 min). Vasodilation responses to the cumulative effect of safranal were normalized as a percentage of the maximum force achieved by 5 × 10^−5^ Phe in each particular aortic segment. 

### 2.5. Effect of Safranal on Phenylephrine-Induced Contraction 

Endothelium-intact aortic rings were incubated with 10^−3^ M safranal for 30 min or with its solvent, and then concentration–response curves to Phe (10^−9^–10^−4^ M) were established. The effect of safranal on the concentration–response curves of Phe was assessed by comparing the maximum contraction induced by Phe and the EC_50_. The contractile responses to Phe were normalized as a percent of the maximum contraction induced by 3 × 10^−3^ M BaCl_2_ that was added at the end of the experiment [27].

### 2.6. Effect of Safranal on Ca^2+^ Influx and Ca^2+^-Induced Contraction 

After one hour of equilibrium for endothelium-intact aortic rings, Krebs bicarbonate buffer was replaced by Ca^2+^-free depolarizing buffer, which had the same composition of Krebs bicarbonate buffer, except NaCl had been replaced isosmotically with KCl, which was increased to 122.7 mM. Tissues were incubated in Ca^2+^-free depolarizing buffer for 30 min, and CaCl_2_ concentration–response curves in the absence and the presence of safranal concentrations (10^−4^ and 10^−3^ M) were established. When the response to the highest concentration of CaCl_2_ reached a steady state, aortic rings were washed with Krebs buffer and contracted by the addition of 3 × 10^−3^ M BaCl_2_. The contractile responses to CaCl_2_ were normalized as a percent of the maximum contraction induced by 3 × 10^−3^ M BaCl_2_ that was added at the end of the experiment [27]. 

### 2.7. Effect of Safranal on Intracellularly Stored Ca^2+^

Another set of experiments with endothelium-intact aortic rings was performed to evaluate safranal’s effect on the release of Ca^2+^ from intracellular stores, and its effect on Ca^2+^ binding to Ca^2+^ receptor proteins of the contractile system. Tissues were incubated for 25 min in a Ca^2+^-free Krebs bicarbonate buffer to which 2 × 10^−3^ M EGTA had been added 30 min after the addition of Ca^2+^-free buffer. Tissues were incubated with safranal (10^−4^, 3 × 10^−4^ or 10^−3^ M) for 15 min and a single dose of 5 × 10^−5^ M Phe was added to obtain the phasic contraction in the absence of an external source of calcium. Tissues were then exposed to a single dose of 3 × 10^−2^ BaCl_2_ in the presence of safranal. When the contraction to BaCl_2_ reached its maximum, aortic rings were washed repeatedly with Ca^2+^-free Krebs bicarbonate buffer to which 2 × 10^−3^ M EGTA had been added, then challenged again with 3 × 10^−2^ M BaCl_2_. The contractile responses to Phe in the presence of safranal and the inhibition of contractions to the first challenge with BaCl_2_ were expressed as a percent of the response to the second challenge with BaCl_2_ (a sample experiment is shown in the Results section for clarification) [22,27]. 

### 2.8. Role of Na^+^/Ca^2+^Exchanger in Safranal-Induced Vasodilation 

Endothelium-intact aortic rings were incubated for 2–3 h with 4 × 10^−4^M ouabain to inhibit Na^+^-K^+^ ATPase. This concentration of ouabain was used based on previous experiments [26]. After the contractions induced by ouabain reached a plateau, different concentrations of safranal (10^−4^, 3 × 10^−4^, or 10^−3^ M) were added and the safranal-induced relaxations were quantified. Aortic rings were then washed several times with Krebs buffer and 3 × 10^−3^ M BaCl_2_ was added to obtain the maximum contraction in the absence of safranal. The vasodilatory responses to safranal were normalized as a percent of the maximum contractions obtained with BaCl_2_. 

Another set of experiments was performed to assess whether safranal blocks the Na^+^/Ca^2+^ exchanger. Aortic rings were preincubated with 10^−3^ M of safranal (or the solvent) for 30 min before the addition of 4 × 10^−4^ M ouabain, and a change in tissue tone was observed. Tissues were then washed several times with Krebs buffer and 3 × 10^−3^ M BaCl_2_ was added to obtain the maximum contraction. The contractile responses to ouabain in the presence of safranal were calculated as a percent of maximum BaCl_2_ contraction.

### 2.9. Statistical Analysis

Data are expressed as the means ± S.E.M. Statistical analysis was performed using one-way ANOVA followed by Fisher’s LSD test. The results of different treatments were then plotted as non-linear regression curve fitting using the Graph Pad Prism version 8.00. Differences with *p* < 0.05 were declared statistically significant. 

## 3. Results 

### 3.1. Effect of Safranal on Rat Aorta Rings Precontracted with Phenylephrine 

Figure 1 shows that safranal (10^−4^–3 × 10^−3^ M) induced vasorelaxation in endothelium-intact and endothelium-denuded aortic rings precontracted with 5 × 10^−5^ M Phe. No significant difference between the effect of safranal on endothelium-intact and endothelium-denuded aortic rings was noticed (Table 1). In contrast, endothelium denudation significantly reduced the vasorelaxation induced by acetylcholine, indicating effective mechanical denudation (Table 1).

### 3.2. Effects of Indomethacin, L-NAME, Methylene Blue, and Quinacrine on Safranal-Induced Vasodilation 

Preincubation with indomethacin, L-NAME, methylene blue, or 10^−6^ M and 10^−5^ M quinacrine had no significant effect on safranal-induced vasorelaxation (Figure 2A–D, respectively). The IC_50_ values of safranal in the presence of these inhibitors were (6.2 ± 0.6) × 10^−4^ M, (4.3 ± 0.3) × 10^−4^ M, (3.5 ± 0.8) × 10^−4^ M, (4.8 ± 0.6) × 10^−4^ M, and (3.9 ± 0.7) × 10^−4^ M, respectively, compared to (5.0 ± 0.6) × 10^−4^ M for the control.

### 3.3. Effect of Safranal on Phenylephrine-Induced Contraction

Figure 3 shows that preincubation of aortic rings with 10^−3^ M safranal significantly inhibited the maximum contraction of aortic rings induced by the cumulative addition of Phe (98.7% vs. 57.5%). The EC_50_ values of Phe for control and safranal-treated rings were (4.5 ± 0.5) × 10^−7^M and (1.0 ± 0.27) × 10^−6^ M, respectively (*p* < 0.004). 

### 3.4. Effect of Safranal on Ca^2+^-Induced Contraction in Calcium-Free Depolarizing Solution

In calcium-free depolarizing solution, safranal (3 × 10^−4^, 10^−3^ M) significantly inhibited the maximum contractions induced by CaCl_2_ added cumulatively. The EC_50_ of Ca^2+^ were not significantly affected, with (1.3 ± 0.5) ×1 0^−2^ M, (1.3 ± 0.5) × 10^−2^ M, and (2.3 ± 0.4) × 10^−2^ M for the control, 3 × 10^−4^ M safranal, and 10^−3^ M safranal, respectively (Figure 4). 

### 3.5. Effect of Safranal Pretreatment on the Ca^2+^ Release from Intracellular Stores 

In Ca^2+^-free buffer, 5 × 10^−5^ M Phe caused the aortic rings to elicit phasic contractions with a magnitude of about 25% of maximum BaCl_2_ contractions (Figure 5). Increasing the concentration of safranal did not reduce the magnitude of the phasic contractions induced by phenylephrine in Ca^2+^-free buffer. In contrast, safranal caused concentration-dependent depression of contractions induced by 3 × 10^−2^ M BaCl_2_.

### 3.6. Safranal Relaxed Ouabain-Induced Contractions

Figure 6A shows a sample experiment in which incubation of aortic rings with 4 × 10^−4^ M ouabain caused a slow development of tension that was equal to 76.95 ± 1.74% of BaCl_2_ maximum contraction in 133.0 ± 8.0 min. Safranal relaxed these contractions in a concentration-dependent manner (Figure 6B). When safranal and ouabain were washed, the aortic rings developed normal potent contractions (Figure 6A).

### 3.7. Safranal Inhibited Ouabain-Induced Contractions

In a final set of experiments, aortic rings were incubated with 10^−3^ M safranal for 30 min. When challenged with 4 × 10^−4^ M ouabain, they were unable to develop any significant contractions (10.9 ± 1.4% of BaCl_2_ maximum contraction) (Figure 7A), whereas aortic rings incubated with the vehicle developed significant contractions to ouabain that were equal to 76.9 ± 1.7% of BaCl_2_ maximum contraction (Figure 7B). All tissue rings were still capable of developing contractions in response to BaCl_2_ after wash out of safranal or the vehicle.

## 4. Discussion 

Data from the present study indicate that safranal induced a vasodilatory effect against phenylephrine-induced contractions in rat aortic rings. The IC_50_ values suggest that this compound had a significant vasodilatory effect, although many Ca^2+^ antagonists have lower IC_50_ values by 1–2 orders of magnitude. Furthermore, the vasodilatory effect was similar in both endothelium-intact and endothelium-denuded aortic rings, indicating that endothelium-derived factors are not required to cause safranal vasodilatation (Figure 1). To further confirm if endothelium was involved in vasodilation mechanism induced by safranal, several experiments were performed using enzyme inhibitors and antagonists, namely, L-NAME, indomethacin, quinacrine, and methylene blue. All these experiments confirmed the lack of the role of endothelium in safranal-induced vasodilation in accordance with the findings of Razavi, et al. (2016), which showed that safranal relaxed the aortic rings predominantly through an endothelium-independent mechanism [17].

This study also ruled out the involvement of prostacyclin or other prostaglandins in safranal-induced vasodilation, since indomethacin, a COX inhibitor, did not modify this vasodilation. This is consistent with the findings of Razavi and coworkers (2018), who showed that vasodilation induced by the aqueous ethanolic extract of the whole saffron in isolated aorta was independent of the cyclooxygenase pathway [21]. Other researchers studying the vasodilatory effect of crocin also concluded a lack of significant difference between tissues incubated with indomethacin and their controls [5,28]. Their data, however, indicated that crocin exerts its vasodilatory effect via induction of nitric oxide production or activation of calcium-activated potassium channels (KCa 3.1), but not PGI_2_. 

In contrast to this latter finding, the current study showed that pretreatment of rat aortic rings with L-NAME, an NOS inhibitor, did not reduce the vasodilatory effect of safranal, indicating that it was not attributed to nitric oxide release (Figure 2B). This chemical “denudation”, along with the mechanical denudation described in Figure 1, allowed us to conclude that nitric oxide has no significant contribution to safranal vasodilation of aortic rings. 

Further support of the absence of a role for nitric oxide in the vasodilatory effect of safranal in the present experiments comes from the experiments with methylene blue, an unspecific guanylate cyclase inhibitor. Although methylene blue is an unspecific inhibitor of guanylate cyclase, it has also been shown to directly inhibit NOS and more potently [25]. These two actions of methylene blue ensure that guanylate cyclase was not involved directly or indirectly in vasodilation. Data in Figure 2C show that vasodilation induced by safranal was not significantly affected by pretreatment with methylene blue, confirming that safranal-induced vasodilation is not mediated by the NO/cGMP pathway. 

Yang, et al. (2018) showed that most endothelial functions depend, to various extents, on changes in [Ca^2+^]_i_ [28]. For example, the COX-mediated PGI_2_ pathway requires phospholipase A_2_ (PLA_2_) activation, which requires a high Ca^2+^ burst resulting from emptying intracellular stores [29]. In the current study, quinacrine, an inhibitor of the PLA_2_ pathway, did not significantly alter the safranal-induced vasodilatory effect when used in two different concentrations, indicating that the PLA_2_ pathway is not involved in the vasodilatory mechanism of safranal (Figure 2D). 

The data from Figure 1 and Figure 2 suggest that safranal-induced vasorelaxation could be attributed to changes in other mechanisms involved in Ca^2+^ metabolism. Therefore, experiments were performed to evaluate the potential antagonistic effect of safranal on phenylephrine- and Ca^2+^-induced contractions. Increasing the cytosolic Ca^2+^ level in vascular smooth muscle can be attributed to the activation of voltage-dependent channels, receptor-operated channels, store-operated channels, as well as the Na^+^/Ca^2+^ exchanger operating in the reverse mode [29]. The binding of the contractile agonist phenylephrine to α-adrenergic receptors in plasma membrane of smooth muscle cells generates inositol-1,4,5-triphosphate (IP3), leading to the release of calcium from the sarcoplasmic reticulum, activation of the myofilaments, and contraction [29,30,31]. In the current study, pretreatment with safranal (1 × 10^−3^ M) significantly (*p* < 0.004) inhibited the contractions induced by the selective adrenoceptor agonist phenylephrine (Figure 3). This observation indicated that safranal interferes with Ca^2+^ influx induced by Phe. This conclusion was supported by the finding that safranal (1 × 10^−4^ and 1 × 10^−3^ M) significantly (*p* < 0.01) suppressed the contractile responses induced by CaCl_2_ in calcium-free depolarizing solution (Figure 4). This observation indicates that safranal attenuated the contractile responses by inhibiting influx of calcium from the extracellular environment either through L-type or through store-operated entry channels in aortic smooth muscle, since the main Ca^2+^ source in such experiments is the extracellular environment [27]. The possibility that safranal inhibited the voltage-gated Ca^2+^ channels is very likely since we found that safranal (10^−4^–3 × 10^−3^M) dose-dependently inhibited contractions induced by 60 mM KCl (Appendix A). 

In another set of experiments with Phe in Ca^2+^-free buffer, EGTA was used to buffer calcium retained within the isolated aortic rings prior to the addition of safranal. When the aortic rings were challenged with Phe under such conditions, the magnitude of phenylephrine-induced contractions remained unchanged as we increased safranal concentrations (Figure 5), indicating that safranal did not affect the release of Ca^2+^ from intracellular stores and presumably Ca^2+^ binding to intracellular receptor proteins. The contractions induced by Phe under such conditions are phasic (Figure 5; inset) since they return to the baseline quickly, presumably because Ca^2+^ released from the SR in a Ca^2+^-free medium will be extruded or lost to the extracellular space, especially in the presence of EGTA [32]. In the same experiment, the contractile responses of the aortic rings to extracellularly added BaCl_2_ were reduced in a concentration-dependent manner in the presence of safranal (Figure 5), suggesting that safranal blocked the influx of extracellular calcium since barium blocks inwardly rectifying K^+^ channels, thus leading to depolarization and contraction [33], and it was found to replace calcium in calmodulin-dependent contractions of skinned smooth muscles of rabbit renal arteries [34]. 

To explore the role of the Na^+^/Ca^2+^ exchange mechanism in safranal-induced vasodilation, ouabain was used to inhibit Na^+^/K^+^-ATPase activity. Inhibition of Na^+^/K^+^-ATPase leads to the accumulation of Na^+^ intracellularly, most probably in a restricted sub-plasmalemmal space colocalized with the high ouabain affinity Na^+^/K^+^-ATPase and with the superficial SR and SERCA in smooth muscle [35,36,37,38]. This ouabain action activated the Na^+^/Ca^2+^ exchanger to work in the reverse mode bringing in Ca^2+^ into the cell and causing a significant tonic contraction in vascular smooth muscle cells (Figure 6). This contraction can be explained according to the lag hypothesis of the action of cardiotonic steroids such as ouabain [39,40,41]. In the present experiments, safranal relaxed, in a concentration-dependent manner, the contractions induced by ouabain. This relaxation by safranal could be due to (i) inhibition of Ca^2+^ influx from the extracellular environment through L-type or store-operated channels, which is needed to maintain the tone in the presence of active SR Ca^2+^-ATPase or plasma membrane Ca^2+^-ATPase; (ii) inhibition of the function of the Na^+^/ Ca^2+^ exchanger, which mobilizes Ca^2+^ into the cell in exchange for Na^+^ exiting the cell; (iii) efflux of K^+^ ions out of the cell through Ca^2+^-activated potassium channels, leading to a relaxation; and (iv) activation of SR Ca^2+^-ATPase or plasma membrane Ca^2+^-ATPase. The first possibility gains support from the experiments in which safranal inhibited Phe-induced, Ca^2+^-induced, and K^+^-induced contractions. The second possibility gains support from the experiments that showed that incubation of aortic rings with safranal followed by challenge with ouabain prevented these preparations from developing any significant contractions (Figure 7). In contrast, the control preparations incubated with the safranal solvent were still able to develop the normal magnitude of contraction. The third possibility is unlikely since the application of ouabain, a situation that blocks the Na-K pump, thus setting the stage for unopposed K^+^ efflux, did not reduce the tonic contractions induced by ouabain that were maintained for hours without decline (time to peak of contraction was 133 ± 8 min). The fourth possibility cannot be ruled out based on the present experiments, and this is one of the limitations of this study. Another limitation is that inhibition of the Na-K pump by ouabain could raise extracellular K^+^, as found by other investigators in hippocampal neurons [42], hence leading to depolarization and to consequences such as the activation of inwardly rectifying K^+^ channels. The concentration of ouabain used in the present experiments may not completely inhibit all the Na-K pump population to cause the accumulation of high K^+^ to cause that effect, since only a 10x increase in ouabain induced a phasic contraction that decayed in about 3 min, and then was followed by a modest relaxation (Appendix A). 

The findings of this study suggest that safranal inhibited the function of the Na^+^/Ca^2+^ exchanger in addition to its inhibitory effect on the voltage-gated Ca^2+^ channels. Although a more direct evidence is needed to substantiate this conclusion, other mechanisms for the vasodilatory effect of safranal are still possible, including a possible effect on voltage-gated K^+^ channels. Therefore, further investigations are needed to elucidate the mechanism of action of safranal, taking into account that (1) most of the available Na^+^/Ca^2+^ exchanger inhibitors hitherto are lacking selectivity since they interfere with other ionic currents such as L-type Ca^2+^ current and K^+^ (I_Kir_) currents. Only recently, some more specific Na^+^/Ca^2+^ exchanger inhibitors were reported, and they are in an experimental stage [43,44]. (2) Safranal could be evaluated for use as a potential Na^+^/Ca^2+^ exchanger inhibitor under some pathological conditions where the exchanger, operating in the reverse mode, induces calcium overload in the vasculature or the heart, and hence may increase vasoconstriction or cardiac contractility. (3) Safranal consumption (i.e., as spice and food additive) could interfere with the action of endogenous ouabain function, which has been implicated in hypertension [40]. (4) Safranal could also interfere with the action of cardiotonic steroids for patients who take these as drugs. (5) Safranal could be developed to reduce the potential toxicity of cardiac glycoside drugs, especially with the reported low therapeutic index of those drugs [39,45].

## 5. Conclusions

The data presented here show that safranal relaxed rat thoracic aorta by an endothelium-independent mechanism. This vasorelaxation was not affected by the inhibition of nitric oxide synthase, cyclooxygenase, guanylate cyclase, or phospholipase A2. In contrast, the vasorelaxant effect of safranal involved blockade of extracellular calcium influx through voltage-gated Ca^2+^ channels, but also involved inhibition of the function of the Na^+^/ Ca^2+^ exchanger. In light of the lack of specific inhibitors for the Na^+^/Ca^2+^ exchanger, the use of safranal under certain pathological conditions, especially when the Na^+^/Ca^2+^ exchanger works in the reverse mode, could be assessed.

## Figures and Tables

**Figure 1 molecules-27-04228-f001:**
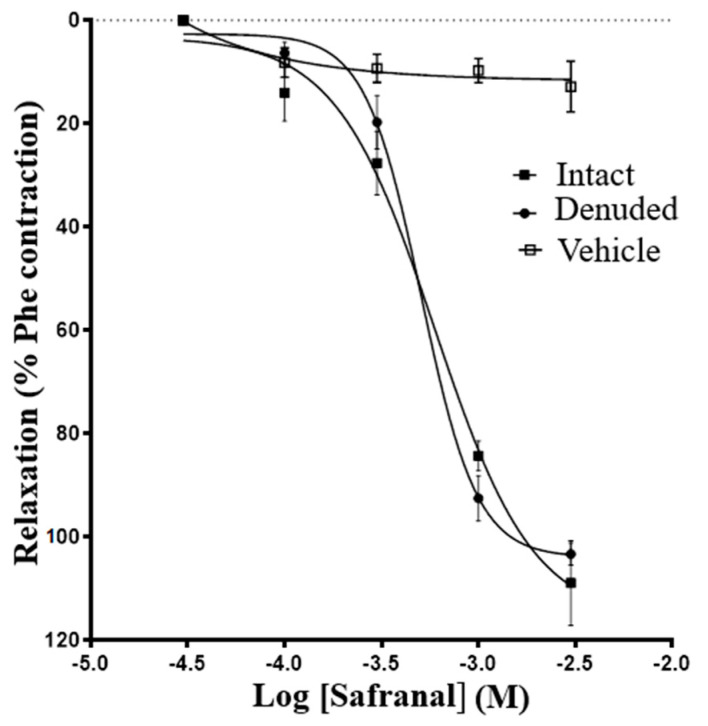
Safranal-induced vasorelaxation in endothelium-intact and endothelium-denuded aortic rings precontracted with phenylephrine. The vasodilatory effect of safranal is expressed as a percentage of vasodilation to maximum contraction induced by 5 × 10^−5^ M Phe in the intact endothelium (solid circles, *n* = 14) and the denuded endothelium (solid squares) rings. The solvent (Tween 80, open circles, *n* = 10) caused negligible vasorelaxation. Values are expressed as means ± SEM.

**Figure 2 molecules-27-04228-f002:**
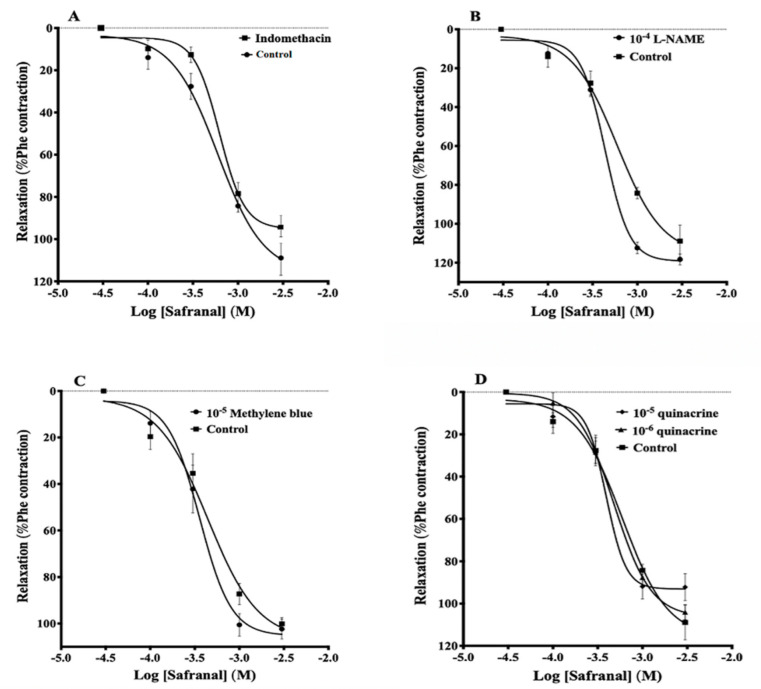
Safranal-induced vasorelaxation in the presence of indomethacin (**A**), L-NAME (**B**), methylene blue (**C**), and quinacrine (**D**). Aortic vessels were precontracted with 5 × 10^−5^ M Phe, then challenged with safranal in the absence or the presence of the inhibitors. Values are expressed as means ± SEM. *n* = 9 for (**A**,**B**); *n* = 7 for (**C**,**D**).

**Figure 3 molecules-27-04228-f003:**
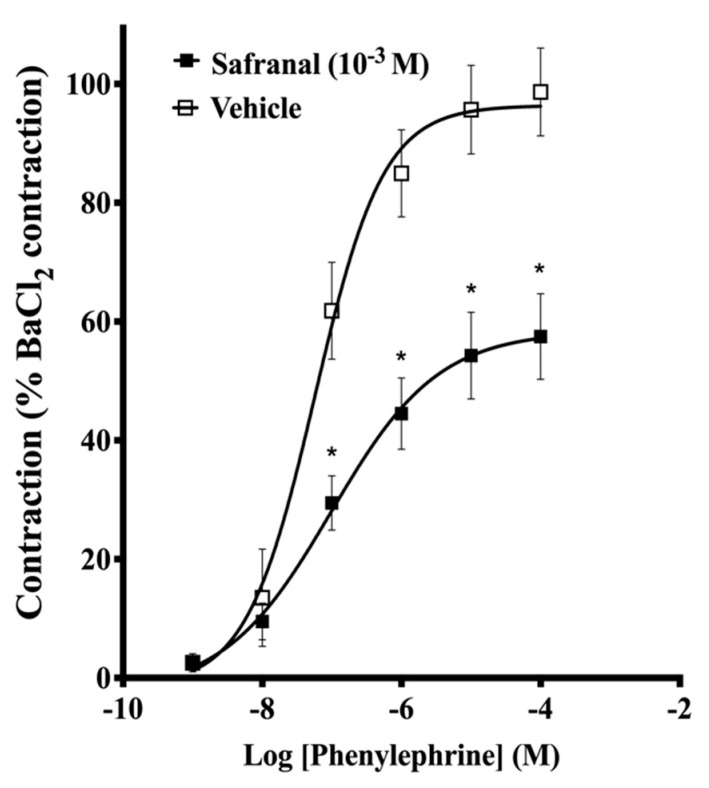
Effect of 10^−3^ M safranal on the concentration–response curves of Phe (10^−9^–10^−4^ M) in aortic rings. Contractile responses are expressed as the percentage of maximum contraction evoked by BaCl_2_ (3 × 10^−3^ M). Data are the means ± SEM of 6 experiments. * indicates significant difference at *p* < 0.05 compared to the vehicle.

**Figure 4 molecules-27-04228-f004:**
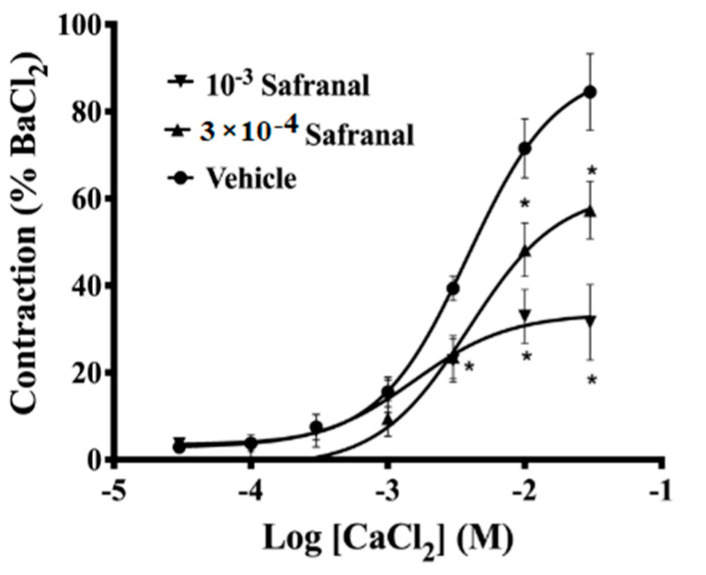
Concentration–response curves of CaCl_2_ (3 × 10^−5^–3 × 10^−2^ M) in the presence of 3 × 10^−4^ M safranal (triangles, *n* = 11) and 10^−3^ M safranal (inverted triangles, *n* = 11) or the vehicle (circles, *n* = 9). Contractile responses are expressed as a percentage of maximum contraction evoked by BaCl_2_ (3 × 10^−3^ M). Values are means ± SEM. * indicates that *p* < 0.05 compared to the vehicle.

**Figure 5 molecules-27-04228-f005:**
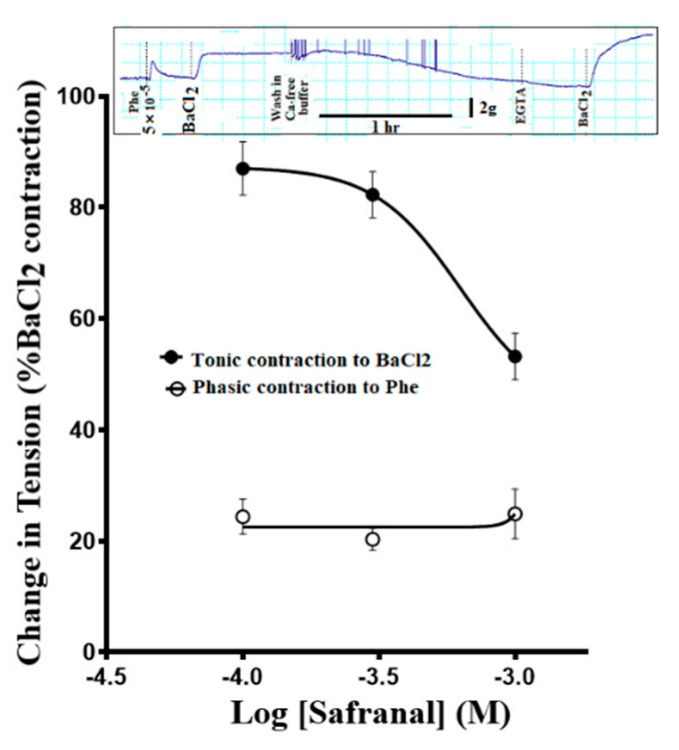
The phasic contractions induced by phenylephrine in a Ca^2+-^free buffer (open circles) are not affected by safranal, whereas contractions induced by 3 × 10^−2^ M BaCl_2_ (closed circles) are inhibited by increasing safranal concentration. Inset in the upper part of the figure represents a sample experiment for clarification. *n* = 6.

**Figure 6 molecules-27-04228-f006:**
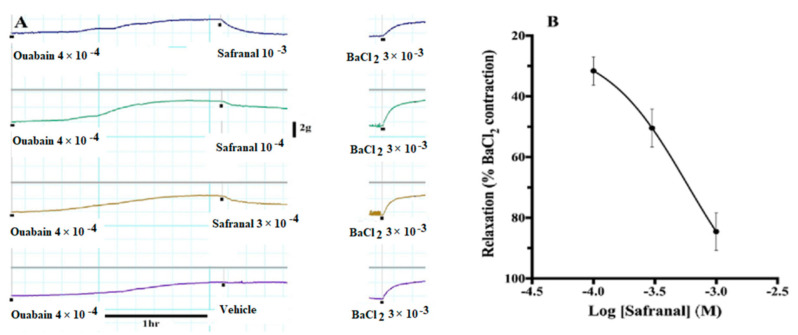
(**A**) Sample experiment in which incubation of aortic rings with 4 × 10^−4^ M ouabain causes the development of sustained contractions over a period of about 2 h. These contractions were relaxed by the indicated concentrations of safranal. BaCl_2_ in the presence of safranal caused modest contractions but when the tissues were then washed with fresh buffer and challenged with 3 × 10^−3^ M BaCl_2_ again, they developed significant contractions. (**B**). The vasorelaxation of ouabain-induced contractions were expressed as a percent of the BaCl_2_-induced contraction. Values are means ± SEM of 6 experiments.

**Figure 7 molecules-27-04228-f007:**
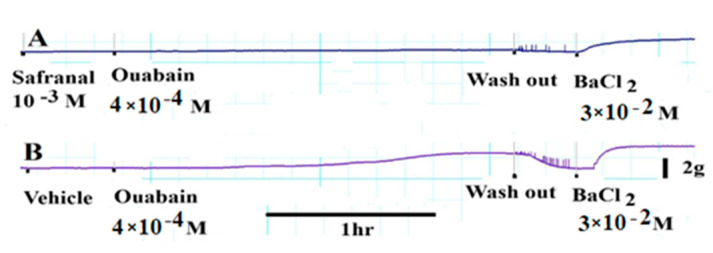
Incubation of aortic rings with 10^−3^ M safranal (**A**) totally blocked the development of tension due to 4 × 10^−4^ M ouabain. Incubation of aortic rings with 4 × 10^−4^ M ouabain causes the development of sustained contractions in the presence of safranal solvent (**B**). *n* = 6.

**Table 1 molecules-27-04228-t001:** IC_50_ for safranal and acetylcholine on denuded and intact isolated thoracic aortic rings ^a^.

Relaxant Agent	Treatment	*n*	IC_50_ (M)	Max. Relaxation (%Phe)
Safranal	Endothelium denuded	10	(6 ± 0.6) × 10^−4^	103 ± 2.1
	Endothelium intact	14	(5 ± 0.6) × 10^−4^	108 ± 8.2
Acetylcholine	Endothelium denuded	6	(3.0 ± 0.3) × 10^−4^ *	27.4 ± 5.2
	Endothelium intact	5	(1.0± 0.7) × 10^−4^	76.2 ± 6.0

^a^ Data are presented as means ± SEM and were obtained using Graph Pad Prism version 8.00. * *p* < 0.02.

## Data Availability

All data generated or analyzed during this study are included in this article. Further inquiries can be directed to the corresponding author.

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
