# Peer review of "Safranal Induces Vasorelaxation by Inhibiting Ca2+ Influx and Na+/Ca2+ Exchanger in Isolated Rat Aortic Rings"

_molecules, 2022, doi:10.3390/molecules27134228_

Round 1

Reviewer 1 Report

Minor Comments:

  • standardize the font in the text, some sentences are with a smaller font
  • It is suggested to shorten the discussion by eliminating the parts concerning saffron or its component crocin since the study concerns safran. In this regard, in the discussion the parts to be deleted are highlighted in yellow.
  • In discussion, Please explain better or deleting this sentence, “ The lack of role for nitric oxide, prostaglandins, cGMP, or PLA2 in safranal-induced vasorelaxation suggested a direct role on Ca2+ metabolism”  which seems to be in contradiction with the one said before. Because Ca2+  plays a key role in the production of NO and PGI2, the fact that NO & PGI2  are not involved would lead to the conclusion that Ca2+ is also not involved.

Author Response

  • standardize the font in the text, some sentences are with a smaller font
  • Response: Thank you for bringing this mistake to our attention. We have standardized the font all through the ms.
  • It is suggested to shorten the discussion by eliminating the parts concerning saffron or its component crocin since the study concerns safran. In this regard, in the discussion the parts to be deleted are highlighted in yellow.
  • Response: We have shortened both the Introduction and the Discussion sections by deleting most of the material related to saffron and to crocin. In the Introduction, we deleted lines 71-75 and lines 84-91 that deal with saffron. In the Discussion section, we deleted lines 296-304 and lines 316-326. These also were dealing with saffron.
  • In discussion, Please explain better or deleting this sentence, “ The lack of role for nitric oxide, prostaglandins, cGMP, or PLA2 in safranal-induced vasorelaxation suggested a direct role on Ca2+ metabolism”  which seems to be in contradiction with the one said before. Because Ca2+  plays a key role in the production of NO and PGI2, the fact that NO & PGI2  are not involved would lead to the conclusion that Ca2+ is also not involved.
  • Response: Thank you for bringing this note to our observation. We have removed that controversial sentence and replaced it with a new text on lines 339-341 of the old version of ms, or 312-313 of the clean version of ms.

Reviewer 2 Report

In this study, the investigators attempted to examine effect of safranal on the vasorelaxant force in isolated aortic rings. There are several comments shown below.

[1] Safranal-mediated vasodilation could be mediated by inhibition of calcium influx from exterior. Should the investigators propose that safranal-mediated effect is due to inhibition of voltage-gated calcium current in vascular smooth muscle. Perhaps, it would be necessary to test whether vascular contraction induced by high-potassium (around 50-70 mM) could be attenuated by adding safranal.

[2] How did the investigators exclude the possibility that safranal-induced vasodilation is mediated through the efflux of potassium ions out of the cell (e.g., activation of calcium-activated potassium currents in vascular endothelial or smooth muscle cells)?

[3] Ouabain, known to an inhibitor of Na/K pump, could secondarily cause the increase sodium/calcium exchanger, thereby increasing calcium influx. On the other hand, the increased calcium influx from the exterior could also be due to the increase in the magnitude of voltage-gated calcium current. Perhaps, either nimodipine, a blocker of L-type calcium current, or BAY K-8644, an activator of L-type calcium current, needs to be pretreated and safranal effect on vascular stripes was further examined. Additional measurements could be needed to verify safranal-mediated vasodilation.

[4] Additionally, extracellular potassium could also be raised by inhibition of Na/K pump, hence leading to membrane hyperpolarization of vascular muscle cells through activation of inwardly rectifying potassium channels. The study with respect to the application of barium chloride could be interesting; however, this agent can effectively block the magnitude of inward rectifying potassium currents. How did safranal affect the endothelial hyperpolarizing factor?

[5] Please describe IC50 values required from safranal-induced vasodilation in the Abstract and Discussion section of the revised manuscript. It is noted from Figure 1 that the IC50 of safranal could be quite high (around 400-500 mM). The value is unlikely to be pharmacologically achievable.

[6] In page 10, in the discussion section, line 4, please carefully use “EC50”. The sentence needs to be appropriately revised for clarity. EC50 should be used for phenylephrine-stimulated effect, not for safranal action.

[7] In page 10, in the second paragraph in the Discussion section, please revise the paragraph. Please state whether safranal-mediated vasodilation is dependent on aortic rings with intact endothelium or can even be accentuated.

[8] Most of descriptions or statements shown in the Discussion section of the manuscript appear to be either over-emphasized or irrelevant to the experimental results. The discussion section in the manuscript needs to be significantly revised.

Author Response

[1] Safranal-mediated vasodilation could be mediated by inhibition of calcium influx from exterior. Should the investigators propose that safranal-mediated effect is due to inhibition of voltage-gated calcium current in vascular smooth muscle. Perhaps, it would be necessary to test whether vascular contraction induced by high-potassium (around 50-70 mM) could be attenuated by adding safranal.

Response: Thank you for stressing this point. Actually, we have suggested in the old version of ms that part of the relaxant effect of safranal was due to inhibition of Ca2+ influx but we did not specify the voltage-gated Ca2+ channels. This was mentioned in the Title of ms and in the concluding statement of the Abstract as well as in the Discussion (Page 11, lines 349-354) and the Conclusion (Page 13, lines 414-415). In the revised version we have specified the voltage-gated Ca2+ channels in the abstract, discussion and conclusion. As a matter of fact, we have gone by your recommendation and looked at the effect of safranal on KCl-induced contraction and we provided (Supplementary Figure 2, attached to your courtesy), to support your point of view and our previous conclusion about inhibition by safranal of Ca2+ influx. 

[2] How did the investigators exclude the possibility that safranal-induced vasodilation is mediated through the efflux of potassium ions out of the cell (e.g., activation of calcium-activated potassium currents in vascular endothelial or smooth muscle cells)?

Response: Thank you for bringing this to our attention. We have included in the revised ms this possibility (page 12 line 380) as one of the possibilities and we discussed its likelihood on page 12 lines 386-389.

 [3] Ouabain, known to an inhibitor of Na/K pump, could secondarily cause the increase sodium/calcium exchanger, thereby increasing calcium influx. On the other hand, the increased calcium influx from the exterior could also be due to the increase in the magnitude of voltage-gated calcium current. Perhaps, either nimodipine, a blocker of L-type calcium current, or BAY K-8644, an activator of L-type calcium current, needs to be pretreated and safranal effect on vascular stripes was further examined. Additional measurements could be needed to verify safranal-mediated vasodilation.

Response: We have thought of such logical experiment. The problem in using nimodipine or nifidipine or other blockes of Ca2+ currents is that they will block the effect of the contractile agent (e.g., phenylephrine) and therefore we will not be able to examine relaxant effect of safranal. As for the activator BAY 8644, we unfortunately had a problem of procuring it, but it would be interesting to test that. 

[4] Additionally, extracellular potassium could also be raised by inhibition of Na/K pump, hence leading to membrane hyperpolarization of vascular muscle cells through activation of inwardly rectifying potassium channels. The study with respect to the application of barium chloride could be interesting; however, this agent can effectively block the magnitude of inward rectifying potassium currents. How did safranal affect the endothelial hyperpolarizing factor?

Response: We have discussed this possibility on page 12 lines 390-395 and we have provided Supplementary Figure 3 (attached) trying to explain a complex action of ouabain. We think that the possibility suggested in point 4 above may occur only when ouabain is used in large concentrations that inhibit a large population of Na/K channels.

 5] Please describe IC50 values required from safranal-induced vasodilation in the Abstract and Discussion section of the revised manuscript. It is noted from Figure 1 that the IC50 of safranal could be quite high (around 400-500 mM). The value is unlikely to be pharmacologically achievable.

Response: Thank you for bringing this to our attention. We have corrected EC50 for safranal to IC50 in Table 1 as well as in the result section (P 7 line 225) and in the discussion (P10 line 288). 

As for the remark on the IC50 of safranal approximated from Figure 1, The real values of IC50 are actually the one shown in Table 1 which is about 0.6 mM.

[6] In page 10, in the discussion section, line 4, please carefully use “EC50”. The sentence needs to be appropriately revised for clarity. EC50 should be used for phenylephrine-stimulated effect, not for safranal action.

Response: we have taken care of that as mentioned in our response to note 5.

[7] In page 10, in the second paragraph in the Discussion section, please revise the paragraph. Please state whether safranal-mediated vasodilation is dependent on aortic rings with intact endothelium or can even be accentuated

Response: We have revised that paragraph and deleted lines 296-302.

[8] Most of descriptions or statements shown in the Discussion section of the manuscript appear to be either over-emphasized or irrelevant to the experimental results. The discussion section in the manuscript needs to be significantly revised.

Response: We hope that we have modified all the bumpy or over-emphasized and irrelevant statements. We thank the referee for taking the time to read our work and to bring many new insights into the discussion.

Note: I have tried to attach 2 supplementary files but I am not sure if they get attached. 

Reviewer 3 Report

This manuscript describes studies directed at trying to define the mechanism of relaxation of phenylephrine pre-contracted rat aortic rings to safranal.  

The major concern relates to the interpretation of the ouabain experiments. In addition to activation of Na+/Ca2+ exchange, ouabain will depolarize the smooth muscle resulting in activation of voltage-gated Ca2+ channels.  Based on your other experiments, it seems likely that safranal -induced inhibition of ouabain-induced contractions involves inhibition of voltage-gated Ca2+ channels.  Your experiments do not allow you to distinguish between effects on Na+/Ca2+ exchange and effects on voltage-gated Ca2+ channels.  Therefore the discussion should be revised accordingly.

Additional comments:

Section 3.2 - How do you know that the concentrations of antagonists used were effective. Usually the effects, for example, of L-NAME on acetylcholine-induced relaxation are shown to verify NOS inhibition.  At the least the lack of control experiments to demonstrate efficacy of these inhibitors should be cited as a limitation of your studies.

page 12, line 20 to page 13, line 6 - See major concern cited above.

Author Response

The major concern relates to the interpretation of the ouabain experiments. In addition to activation of Na+/Ca2+ exchange, ouabain will depolarize the smooth muscle resulting in activation of voltage-gated Ca2+ channels.  Based on your other experiments, it seems likely that safranal -induced inhibition of ouabain-induced contractions involves inhibition of voltage-gated Ca2+ channels.  Your experiments do not allow you to distinguish between effects on Na+/Ca2+ exchange and effects on voltage-gated Ca2+ channels.  Therefore the discussion should be revised accordingly.

Response: Thank you for reminding us with this point. What you suggested (i.e. safranal inhibition of voltage gated Ca2+ channels) is very likely since we have shown in Figures 3 and 4 in the old version of ms that safranal inhibited both phenylephrine-induced and Ca2+-induced contraction. Both of these observations are resulting from inhibition of Ca2+ influx through voltage-gated Ca2+ channels. This conclusion has been mentioned in the title of ms and in the concluding statement in the abstract as well as in the discussion and conclusion. The only thing that we did not stress is to mention the type of channel through which Ca2+ influx occurs. Now in the revised ms we have stressed that in the abstract (line 30) as well as in the discussion. Furthermore, to substantiate this conclusion, we also did safranal-dose-response curve on aortas precontracted with KCl and it confirmed your suggestion (supplementary Figure 2; attached to your courtesy). This confirmation was mentioned again in the Discussion (page 10 line 328).

As for the role of safranal on Na/Ca exchanger: Since safranal did not block completely contractions induced by phenylephrine or by Ca2+, we thought there is another component of Ca2+ influx that safranal might work on. This component can be studied when Na/K pump is inhibited with ouabain which we found to cause well-maintained contraction resulting from the Na/Ca exchanger working in the reverse mode. This ouabain-induced contraction was both relaxed by safranal but also would not develop when aortas were pretreated with safranal.

Section 3.2 - How do you know that the concentrations of antagonists used were effective? Usually the effects, for example, of L-NAME on acetylcholine-induced relaxation are shown to verify NOS inhibition.  At the least the lack of control experiments to demonstrate efficacy of these inhibitors should be cited as a limitation of your studies.

Response: In our experiment we used 10-4M of L-NAME guided by data from literature. The group of S. Moncada in 1990 has found that actually 10uM of L-NAME inhibited by about 98% acetylcholine-induced relaxations of rat aorta. As a precaution we used (as many paper cited) a 10-fold larger concentration. We have included the paper by Moncada group in the revised copy (reference 24). The concentrations of other inhibitors have been also chosen depending on other researchers use which we cited as references 23, 25, and 26. 

Round 2

Reviewer 2 Report

There are several queries that remained, although the investigators answered some of questions raised previously. The queries are shown below.

[1] Safranal could not be used for studies on Na+-Ca2+ exchanging process, given that it can directly suppress the magnitude of voltage-gated Ca2+ currents.

[2] The results showed the ability of safranal to suppress ouabain-induced increase of contractile force. However, to verify safranal effect on voltage-gated Ca2+ current, the effect of Bay-K-8644, an activator of Ca2+ currents, would be important to test if this compound can reverse safranal-mediated inhibition of contractile force.

[3] Whether vascular contraction induced by high-potassium (around 50-70 mM) could be attenuated by adding safranal is also important to be tested. In line 247, how can the “calcium-free depolarizing solution” be prepared in the experimental protocol?

[4] The last paragraph in the Discussion section of the manuscript is lengthy and needs to be extensively revised. Alternatively, effect of ouabain itself on the contractile force in isolated aortic strips could be rather complicated and warrant further investigations. Safranal-mediated attenuation of ouabain-mediated contraction became even confusing to certain extent.

[5] The IC50 of safranal could be quite high (around 400-500 mM). The value seems unlikely to be of pharmacological relevance. In line 248, EC50 needs to be replaced with IC50.

[6] It is also possible that safranal itself exerts some effects on voltage-gated K+ currents in vascular smooth muscle cells.

Author Response

Dear Reviewer

We thank you for your untiring perseverance in following the minor details of our manuscript. This actually opened our eyes to the many other possibilities that may explain the action of safranal. This perseverance is highly welcomed especially when we see that many reviewers these days are only touching superficially on the manuscripts they referee. Please find below our responses to your comments, hoping that you will find them satisfactory.

[1] Safranal could not be used for studies on Na+-Ca2+ exchanging process, given that it can directly suppress the magnitude of voltage-gated Ca2+ currents.

Response: It is true that safranal inhibited the voltage-gated Ca2+ currents, but in the presence of ouabain in the concentration that we used in the present experiment, the source of Ca2+ available to cause the tonic contraction is that resulting from the function of the Na/Ca exchanger working in the reverse mode. This tonic contraction is characterized by slow kinetics (reaching a peak in 133±8 minutes), which is quite different from that induced by voltage-gated Ca2+ currents which has much faster kinetics (for example, phenylephrine-induced contractions reaching a peak in 25 minutes). In our experiments, we used ouabain to provide the condition to study the effect of safranal on Na/Ca exchange process.

[2] The results showed the ability of safranal to suppress ouabain-induced increase of contractile force. However, to verify safranal effect on voltage-gated Ca2+ current, the effect of Bay-K-8644, an activator of Ca2+ currents, would be important to test if this compound can reverse safranal-mediated inhibition of contractile force.

Response: We have responded to this suggested test in our response to Round 1. It was unfortunate that we could not secure the needed amount of Bay K8644 within the time frame given to submit the revision. We, however, performed other suggested experiments and included them as Supplementary Figures 2 and 3. In general, a single study cannot answer all the questions that comes to the human brain. In fact, it should open more questions. But the suggestion is quite logical.

[3] Whether vascular contraction induced by high-potassium (around 50-70 mM) could be attenuated by adding safranal is also important to be tested. In line 247, how can the “calcium-free depolarizing solution” be prepared in the experimental protocol?

Response: As we stated in our response to Round 1 of your report: We have tested the effect of safranal on contractions induced by 60 mM KCl and we provided the data in the form of a Supplementary Figure 2, and we commented on the results in the Discussion Page 11 lines 360-362.

Ca2+-free depolarizing solution preparation is described in page 4 1ines 161-164: The number 122.7 mM resulted from addition of 118mM (replacing Na) + 4.7 mM KCl =122.7mM.

[4] The last paragraph in the Discussion section of the manuscript is lengthy and needs to be extensively revised. Alternatively, effect of ouabain itself on the contractile force in isolated aortic strips could be rather complicated and warrant further investigations. Safranal-mediated attenuation of ouabain-mediated contraction became even confusing to certain extent.

Response: We agree with you that the effect of ouabain itself on the contractile force in isolated aortic rings is rather complicated and warrants further investigation. We have provided in our response to comments from Round 1 Supplementary Figure 3 which shows the dose-response curve of ouabain on aortic rings. Higher concentrations of ouabain were associated with a transient contraction with fast kinetics (probably resulting from depolarization due to accumulation of high K outside due to inhibition of large population of Na/K pumps. The high accumulation of K has been shown experimentally using K electrodes by others [Reference 42]). Moreover, this phasic contraction was followed by a relaxation. We described these effect in the last paragraph of the Discussion (Page 12 lines 396-401) in response to your previous suggestion that safranal effect could be due to efflux of K+ ions out of the cell through Ca2+ - activated potassium channels. We think we ruled out this possibility, but again the action of ouabain could be really intriguing.

The last paragraph of Discussion has now been divided into 2 paragraphs and the reason why the last part of the Discussion is little lengthy is that we feel that it is educational for some readers, especially graduate students, to consider different aspects to consider when, and if, they study ouabain and/or safranal. This is not a promotion for safranal as a medication; it is rather a suggestion for which aspects could be studied further.

[5] The IC50 of safranal could be quite high (around 400-500 mM). The value seems unlikely to be of pharmacological relevance. In line 248, EC50 needs to be replaced with IC50.

Response: We responded to this comment in Round 1. We do not know how did you reach to this high IC50 value, especially when the highest concentration of safranal that we used was 3mM! I guess you approximated that value by looking at Figure 1. But Figure 1 shows that the approximate IC50 intersects with the value of (-3.5) on the Log scale (the X-axis). The antilog for (-3.5) would be 3x10-4 M, or 0.3 mM, not 400 or 500 mM. At any rate we have specified the IC50 values in Table 1 and they are 0.5 and 0.6 mM for endothelium-intact and endothelium denuded aortae, respectively.

The EC50 values mentioned in line 248 are those for Ca2+ in the absence and the presence of safranal and not for safranal (CaCl2 concentration-effect curves), therefore they were expressed as EC50 and not IC50. Nevertheless, we have gone again over the manuscript to make sure that all values describing the effective concentration of safranal are presented as IC50.

[6] It is also possible that safranal itself exerts some effects on voltage-gated K+ currents in vascular smooth muscle cells.

Response: This possibility has been mentioned in the revised ms (page 12 lines 404-405). Ruling out such possibility may require an electrophysiology approach which requires extending the Materials and Methods section beyond the acceptable limits for a single manuscript.
